# The Muscle Fibre Characteristics and the Meat Quality of *m. longissimus thoracis* from Polish Native Złotnicka Spotted Pigs and the Crossbreed Fatteners from the Crossing of Duroc and Polish Large White Boars

Karolina Szulc [1], Dorota Wojtysiak [2], Łukasz Migdał [2,*] and Władysław Migdał [3]

1. Department of Animal Breeding and Product Quality Assessment, Faculty of Veterinary Medicine and Animal Science, Poznań University of Life Sciences, Złotniki, ul. Słoneczna 1, 62-002 Suchy Las, Poland; karolina.szulc@up.poznan.pl
2. Department of Animal Genetics, Breeding, and Ethology, Faculty of Animal Science, University of Agriculture in Kraków, al. Mickiewicza 24/28, 30-059 Krakow, Poland; dorota.wojtysiak@urk.edu.pl
3. Department of Animal Products Technology, Faculty of Food Technology, University of Agriculture in Kraków, ul. Balicka 122, 31-149 Krakow, Poland; wladyslaw.migdal@urk.edu.pl
* Correspondence: lukasz.migdal@urk.edu.pl

**Abstract:** The aim of the investigations were to assess the meat raw material of the domestic Złotnicka Spotted swine breed as well as its hybrids with Duroc and Polish Large White breeds with respect to quality, technological usefulness, and muscle fibre composition and structure. The suitability of individual swine genetic groups (Złotnicka Spotted × Złotnicka Spotted, Złotnicka Spotted × Duroc, Złotnicka Spotted × Polish Large White, Złotnicka Spotted × Złotnicka Spotted/Duroc) for the production of heavy fatteners which can be used as slaughter raw material to manufacture raw and raw-ripening meat products was ascertained. Złotnicka Spotted pigs were characterised by a statistically significantly smaller proportion of IIB fibres and a higher share of I and IIA fibres in the *longissimus thoracis* muscle in comparison to the hybrids of this breed with Duroc and Polish Large White breeds. The diameter of all muscle fibre types in the *longissimus thoracis* muscle of the Złotnicka Spotted breed was greater than in hybrids. No statistically significant differences were found between the parameters of colour and the free drip and the water absorbability of the loin of the Złotnicka Spotted breed pigs and their hybrids with Duroc and Polish Large White breeds. The examined meat was characterised by a small free drip and good water absorbability. The hybrid pigs (Złotnicka Spotted × Duroc) were characterised by the highest content of intramuscular fat (IMF) in the *longissimus thoracis* muscle, which resulted in the lowest sheer force of roasted loin and the lowest thermal drip compared to other hybrids. Smoked, raw loin obtained from the *longissimus thoracis* muscle of the Złotnicka Spotted breed was found to be the most tender, whereas tenderness of the identical loin derived from the *longissimus thoracis* muscle of the Złotnicka Spotted bred hybrids with the Duroc and Polish Large White breeds was poorer. Due to the small headage of the Złotnicka Spotted breed, the appropriate numbers of fatteners of good meat quality parameters suitable to manufacture raw, ripening meat products can only be secured by the crossing of this breed with other meat breeds. Therefore, it appears that crossing the Złotnicka Spotted pigs with Duroc pigs would be a suitable solution.

**Keywords:** pigs; fatteners; native breed; muscle fibre; meat quality

## 1. Introduction

Apart from highly efficient goods, the market of meat products comprises products of low efficiency which are usually considered as luxury products. The above-mentioned group also includes raw, ripening products characterised by specific, very good organoleptic

properties, namely Parma hams, Iberico hams or Polish raw smoked meat products, e.g., Lublin loin or Podlaski kumpiak. The production of raw-ripening meat requires specific raw material in the form of meat from heavy pigs with slaughter body weights of over 120 kg, intramuscular fat content exceeding 3%, and which should be fed traditionally using extensive methods (breeds whose meat has long been used to manufacture such products include: Iberico, Casertana, Alentejana and Mangalica [1]. Polish breeding is still in possession of swine breeds which were used in the past, and they can still be used for the production of high-quality raw ripening products, although populations of these breeds are small [2,3]. These pigs include the following native breeds: the Złotnicka Spotted and the ZłotnickaWhite and Puławska [3–5]. The Złotnicka Spotted breed has not been improved by crossing with other breeds. Due to its small size, conservation breeding aims to maintain biodiversity, and it is not focused on selection to increase production.

That is why Złotnicka Spotted (ZS) pigs have maintained good meat quality suitable for the manufacture of traditional products [6,7]. This was confirmed by experiments carried out by Buczyński et al. [8]; Kapelański et al. [9], Grześkowiak et al. [10] and Szulc et al. [11], which revealed that the meat of the Złotnicka Spotted breed was characterised by a lack of quality deviations, small free drip and good pH. In accordance with the assumptions of the conservation program of the Złotnicka Spotted breed genetic resources [12], attempts have been made to maintain the useful traits of these pigs at a constant level, while simultaneously maintaining diversity within the population, which prevents increases in the levels of inbreeding. In addition, the realisation of the conservation program of the Złotnicka Spotted breed genetic resources also aims to increase the size of this population to the level commonly considered as safe because the number of sows of the basic herd is nearing 500 individuals. In 2007, the sow herd size of the Złotnicka Spotted breed reached 318 kept in 30 herds, in 2010—1021 sows kept in 46 herds, in 2015—421 sows kept in 21 herds and in 2019—914 sows kept in 30 herds [3]. The specific character of conservative breeding results in low values for fattening and the slaughter performance traits observed in the Złotnicka Spotted breed [11,13]. However, in market economy conditions, maintenance of each animal breed must be economically justified. Due to the small size of this breed as well as the poor results of its fattening and slaughter traits, attempts are being made to use pigs of the Złotnicka Spotted breed in interspecific crossing. Attempts made so far to cross sows of this breed with Pietrain and Hampshire breed boars have failed to produce satisfactory results [14,15]. The meat used to manufacture raw and raw-ripening products must be characterised by a pH of 5.6–5.8, approximately 3.5% intramuscular fat, as well as suitable muscle fibre composition and structure. It is evident from experiments carried out so far that the muscle fibre size and muscle fibre type percentage have a decisive impact on muscle "functional character" and, hence, on its quality as well as its technological and culinary value [16]. Muscle fibres are classified as type I (slow-twitch oxidative), IIA (fast-twitch oxidative glycolytic) and IIB (fast-twitch glycolytic) [17–21]. Muscle fibre characteristics such as fibre density, size or type proportion influence the meat quality traits of pork, including water-holding capacity, colour and texture. Moreover, correlations between proportions of individual types of muscle fibres, the surface occupied by them and meat technological and consumption quality are also recognised [22–27], hence, it is possible to exert influence not only on the muscle functional character but also its quality [16,20,21]. The muscle fibre profile is characteristic for breeds, lines and hybrids [28–31]. According to Wojtysiak and Połtowicz [32], autochthonous breeds have a higher content of oxidative muscle fibres in muscles than modern breeds. In addition, it also depends on the rate of growth and the lean meat content [33]. Gil et al. [34] found that Pietrain and Meishan pigs were characterised by extreme sizes of muscle fibres; Pietrain had fibres of the greatest diameter, while Meishan had the smallest diameter. On the other hand, Large White, Landrace and Duroc pigs were characterised by moderate muscle fibres. Moreover, the Duroc breed is also indicated as a breed characterised by advantageous meat quality features [35–37]. The objective of the current investigation was to evaluate the raw meat material of Złotnicka Spotted breed pigs and their hybrids with Duroc and Polish

Large White breeds with respect to the quality, technological suitability and structure of muscles. The essence of the studies was determination of the suitability of the following individual swine genetic groups: Złotnicka Spotted × Złotnicka Spotted (ZS), Złotnicka Spotted × Polish Large White (ZS × PLW), Złotnicka Spotted × Duroc (ZS × D), Złotnicka Spotted × Złotnicka Spotted/Duroc [(ZS × ZS) × D] for the production of heavy fatteners used as slaughter raw material to manufacture raw and raw-ripening products.

## 2. Materials and Methods

### 2.1. Animals and Diets

The investigation used 50 carcasses of fatteners from the following four genetic groups: ZS—Złotnicka Spotted × Złotnicka Spotted ($n$ = 20)—purebreed fatteners, ZS × PLW—Złotnicka Spotted × Polish Large White ($n$ = 10), ZS × D—Złotnicka Spotted × Duroc ($n$ = 10) and (ZS × ZS) × D—Złotnicka Spotted × Złotnicka Spotted/Duroc ($n$ = 10). The animals were divided into four experimental groups keeping the sex ratio at 1:1. All the animals were tattooed and ear-marked. Experimental animals with the average weight of 20 kg were selected and the experiment was terminated when the animals attained the slaughter weight of approximately 120 kg (113.0–123.6 kg). The experiment was divided into the starter (20–30 kg), grower (30–80 kg) and finisher (over 80 kg) periods. Table 1 shows the approximate composition of diets. The rations in all of the three stages were similar for all of the four crosses. The fatteners were kept in collective pens of 10 animals per pen on shallow bedding. They were fed *ad libitum* with total mixed rations and they had constant access to water. After the final fattening, the animals were transported from the farm to the abattoir located at a distance of approximately 50 km. The slaughter of animals was conducted after two hours' rest. Fatteners were shocked with a current using the Polish apparatus STZ 6 KTM 29.53.16.50 SWW 0782-11, year of production 2015 (P.P.U.H."KOMA" Sp. z o.o. Wilkanowo) of the following parameters: an intensity of approximately 1.5 A, a frequency of 50 Hz, a voltage of 250 V, the time of the was shock 8 s and the time from the shock to the pricking was 20 s. About 30 min after the slaughter, carcasses were weighed with a 100 g accuracy; meat content was estimated with the help of the optical-needle apparatus CGM; and, backfat in the following five measuring points was also assessed: over the shoulder, on the back and on the cross in point I, II and III [38]. Later, half-carcasses were cooled using a mono-gradual system at the temperature of approximately 4 °C.

**Table 1.** Proximate composition of diets.

| Items | Diets | | |
|---|---|---|---|
| | **Starter** | **Grower** | **Finisher** |
| Spring barley (%) | 26.00 | 20.00 | 21.00 |
| Triticale (%) | - | 20.00 | 39.80 |
| Winter wheat (%) | 30.40 | 16.10 | - |
| Maize (%) | 20.00 | 13.00 | 12.00 |
| Soybean meal 46 (%) | 16.00 | 13.00 | 7.40 |
| Rapeseed meal "00" 34 (%) | - | 5.00 | 2.00 |
| Wheat bran (%) | - | 10.00 | 15.00 |
| [1] Ekonomix T (%) | 2.40 | 1.50 | 1.30 |
| Fodder chalk (%) | 1.10 | 1.40 | 1.50 |
| [2] Zinteral (%) | 0.10 | - | - |
| [3] Substimel 950 (%) | 4.00 | - | - |
| Dry matter (%) | 90.29 | 90.28 | 90.56 |
| Energy (MJ/kg) | 13.46 | 12.63 | 12.46 |
| Crude protein (%) | 16.26 | 17.31 | 14.79 |
| Digestible protein (%) | 13.65 | 14.65 | 12.65 |
| Lysine (%) | 1.11 | 0.99 | 0.79 |

[1] Mineral feeding stuff, [2] Additive of Zn, [3] Additive of whe.

## 2.2. Raw and Roasted Meat

After 24 h of cool storage meat ageing (single-stage cooling, end temperature 4 °C, humidity of air 88–90%, air velocity 2 m/s), samples of *m. longissimus thoracis* (LT) were collected from half-carcasses. The meat was cut into slices: half of them were further analysed as raw meat, while the other half was wrapped in aluminium foil and roasted in an electric oven (Amica, type 6226 CE3). The chops were roasted at 180 °C to reach an internal temperature (thermometer type DT-34, Termoprodukt, Poland) of 78 °C and then cooled to room temperature (30 min), stored for 45 min in a 4 °C cooler and weighed for cooking loss determinationat. For measurement of shear force, five cylindrical specimens (14 mm diameter and 15 mm high) were cut from the raw and toasted LT slices with a hand-held coring device (cork bore).

## 2.3. Raw Smoked Loin

Raw, smoked loin was prepared from the *longissimus thoracis muscle* (LT) using traditional technology, i.e., without functional additives. The cleaned, lean meat was pickled in brine with the following composition: table salt—6%, sodium nitrite—0.055%, sugar—1.1%, sodium ascorbate—0.4% and water—92.805%. After smoking, the pH (pH meter CP-46 with a dagger electrode ERH-12-6N, ELMETRON, Zabrze, Poland), colour and tenderness (with the assistance of the Warner–Blatzler apparatus type TA.XT.plus, Stable Micro Systems, Co., Ltd., Godalming, UK) of the obtained product were determined. The meat colour was determined using a Konica Minolta CM–600d spectrophotometer (Minolta Co., Ltd., Tokyo, Japan) with a 50-mm diameter measuring head in the CIE L*a*b* system, where the L* parameter corresponds to the degree of lightness (L* = 0:black, L* = 100:white) and a* and b* are colour components (a* > 0 red, a* < 0 green, b* > 0 yellow, b* < 0 blue). The chromametre was calibrated against a white plate standard (Y = 93.8, x = 0.3136, y = 0.3192). The texture profile (TPA) of the meat was analysed according to the PN-ISO Norm 11,036:1999 with the TA.XT.plus texture analyser (Stable Micro Systems Co., Ltd., Godalming, UK). Shear force was measured from cylindrical samples (14 mm diameter, 15 mm height) using a Warner–Bratzler attachment (shearing blade thickness of 1.016 mm, V-shaped cutting blade with a 60° angle, corner of the V rounded to a quarter-round of a 2.363 mm diameter circle, spacers providing the gap for the cutting blade to slide of 1.245 mm thickness) and a triangular notch in the blade. The blade speed during the test was 1.5 mm/s. The results are presented as force per area ($kg/cm^2$).

## 2.4. Muscle Fibre

For histochemical examinations, muscle samples were obtained 30 min *postmortem* from the right side of the carcasses, deep within the *longissimus thoracis* (LT) muscle. Muscle samples were cut into 1 $cm^3$ pieces (parallel to the muscle fibres) and frozen in isopentane that was cooled using liquid nitrogen and stored at −80 °C until subsequent analyses. The samples were mounted on a cryostat chuck with a few drops of tissue-freezing medium (Tissue-Tek; Sakura Finetek Europe, Zoeterwoude, The Netherlands). Transverse sections (10-μm thick) were cut at −20 °C in a cryostat (Slee MEV, Nieder-Olm, Germany). To distinguish muscle fibre types (I, IIA and IIB), a modified combined method of NADH-tetrazolium uctase activity was used and immunohistochemical determination of the slow myosin heavy chain on the same section with monoclonal antibodies against the skeletal slow myosin heavy chain was performed for 1 h at RT (NCL-MHCs, clone WB-MHCs Leica Biosystems, Buffalo Grove, IL, USA, dilution 1:80) [39]. The reaction was visualized by the NovoLinkTM Polymer Detection System (Leica Biosystems, Newcastle Upon Tyne, UK) according to the manufacturer's instruction. Finally, all sections were dehydrated in a graded series of ethyl alcohol, cleaned in xylene and mounted in DPX mounting medium (Fluka, Buchs, Switzerland). A minimum of 300 fibres were counted in each section using a NIKON E600 light microscope. The percentage and diameter of muscle fibre types were quantified with an image analysis system using the Multi Scan v. 14.02 (Computer Scanning Systems Ltd., Warsaw, Poland) computer software.

## 2.5. Meat Quality

At 45 min ($pH_{45}$) and 24 h ($pH_{24}$) after the slaughter, pH was measured in the *longissimus thoracis* (LT) muscle at the level of 11th thoracic vertebra by means of a pH meter CP-46 with a dagger electrode ERH-12-6N, (ELMETRON, Zabrze, Poland). Samples from the *longissimus thoracis* (LT) muscle were collected for laboratory investigations. The following measurements were made of the raw meat samples:

- Water content according to the standard PN-ISO 1442:2000 [40];
- Fat content according to the standard (PN-ISO 1444:2000) by means a Soxtec ST 243 apparatus (FOSS, Hilleroed, Denmark) [41];
- Protein content with the Kjeldahl method (PN-75/A-04018) by means of a Büchi Distillation Unit B-324 apparatus (BÜCHI Labortechnik AG, Flawil, Switzerland) [42];
- Meat weight loss in cooking: the samples were heated to reach the internal temperature of 75 °C in the geometric centre of the sample. The results were computed from the difference between the weight before and after cooking [43];
- The lightness [L*], redness [a*] and yellowness [b*] of the meat was determined using a Konica Minolta CM-600d spectrophotometer (Minolta Co., Ltd., Tokyo, Japan). The values of [a*] and [b*] were used to calculate the saturation value—chroma [C*];
- Drip loss: after measurement of the weight, an approximately 100 g sample of the LT muscle was placed in a plastic bag and stored in a refrigerator at the temperature of 4 °C for 48 h. After that time, the samples were weighed and the results were computed from the weight difference.

## 2.6. Instrumental Measurement of Shear Force

Five cylinder-shaped samples (14 mm in diameter and 15 mm in height) were cut from the meat, roasted in the oven at 180 °C to the internal temperature of 78 °C; then, they were cooled to room temperature (30 min) and stored for 45 min in a 4 °C cooler. The shear force was measured using a TA-XT2 Texture Analyser (Stable Micro Systems Co., Ltd., Godalming, UK) with a Warner–Bratzler attachment and a triangular notch in the blade. The blade speed during the test was 1.5 mm/s. The result was presented as force per area ($kG/cm^2$).

## 2.7. Instrumental Measurement of Texture Parameterspn ISO 11,036: 1999

The five cylinder-shaped samples (14 mm in diameter and 15 mm in height) were cut from meat roasted as above [44]. The texture was analysed using a TA-XT2 Texture Analyser (Stable Micro Systems Co., Ltd., Godalming, UK) with an attachment in the form of a cylinder 50 mm in diameter. The samples were subjected to a double pressing test using a force of 10 g to 70% of their height. The cylinder speed was 2 mm/s and the interval between pressures was 3 s.

## 2.8. Statistical Analysis

Two-way analysis of variance was performed in SAS® v. 9.2. [45].The effects of gender and genetic groups were fitted. The slaughter weight and the slaughter age were included as covariates. Effects that were significant in ANOVA were followed by post-hoc Tukey's test to determine pairs of significantly different means at level 0.05. Additionally, Pearson correlations between analyzed traits were estimated.

## 3. Results and Discussion

Fatteners of the Złotnicka Spotted breed (ZS × ZS) were characterised by the lightest loin (3.61 kg—4.06% carcass weight) in comparison with Duroc (ZSD: 3.83 kg—4.32% carcass weight; (ZS × ZS) × D—3.82 kg—4.22% carcass weight)) and with Polish Large White (3.95 kg—4.53% carcass weight). Szulc et al. [11,13,46,47] confirmed that meat of the Złotnicka Spotted breed is characterised by excellent quality and that crossing ZS and Duroc breeds improves fattening and slaughter performance, while maintaining good meat quality in their crosses. As other economic traits were analysed and showed significant

differences between breeds ([11,13,46,47]), we decided to analyse the structure of muscle fibre for possible explanation of the above-mentioned differences.

Table 2 presents percentage proportions of individual muscle fibres in the *longissimus thoracis* muscle of Złotnicka Spotted breed pigs as well as hybrids of this breed with Duroc and Polish Large White. Złotnicka Spotted pigs were characterised by a statistically significantly smaller proportion of IIB fibres, a greater share of I fibres, and a proportion of IIA fibres in the *longissimus thoracis* muscle in comparison with hybrids with the Duroc and Polish Large White. On the other hand, the diameter of all types of muscle fibres in the *longissimus thoracis* muscle was greater than in hybrids of this breed as shown in Table 3. Investigations of muscle fibre profiles from the *longissimus lumborum* muscle of Złotnicka Spotted breed pigs carried out by Bogucka and Kapelański [48] revealed that muscles constituted only 13.83% of type I muscle fibres, IIA fibres—23.71% and IIB fibres—62.46%. The proportions of IIB fibres in the Złotnicka Spotted breed was the lowest of all the analysed breeds (Polish Landrace, Pietrain, Złotnicka Spotted and triple-breed hybrids of Pietrain/PL/PLW).

**Table 2.** The muscle fibre type percentage (%) of the *m.longissimus thoracis* for male and female fatteners of the Złotnicka Spotted (ZS) breed and the hybrid crosses.

| Genetic Groups of Fatteners | Muscle Fibre Type Percentages | | |
|---|---|---|---|
| | IIB $\bar{x} \pm SD$ | IIA $\bar{x} \pm SD$ | I $\bar{x} \pm SD$ |
| | Breed | | |
| ZS | 57.38 [a] ± 4.92 | 21.57 [a] ± 2.38 | 21.05 [a] ± 4.01 |
| ZS × D | 61.46 [b] ± 1.13 | 19.38 [b] ± 2.31 | 19.62 [a] ± 1.82 |
| (ZS × ZS) × D | 64.94 [c] ± 6.25 | 20.83 [a,b] ± 1.58 | 14.24 [b] ± 6.14 |
| ZS × PLW | 69.45 [d] ± 1.57 | 15.15 [c] ± 1.23 | 15.40 [b] ± 0.95 |
| *p*-value | 0.0008 | 0.0000 | 0.0003 |
| | Barrows | | |
| ZS | 52.80 [a] ± 1.24 | 22.63 [a] ± 0.62 | 24.57 [a] ± 1.70 |
| ZS × D | 61.27 [b] ± 1.08 | 19.92 [b] ± 1.18 | 19.72 [b] ± 1.75 |
| (ZS × ZS) × D | 64.81 [c] ± 5.52 | 20.29 [b] ± 1.52 | 14.20 [c] ± 5.24 |
| ZS × PLW | 68.67 [d,c] ± 1.23 | 15.43 [c] ± 1.46 | 15.90 [c] ± 0.85 |
| *p*-value | 0.001 | 0.0000 | 0.0006 |
| | Gilts | | |
| ZS | 61.96 [a] ± 1.73 | 20.50 [a] ± 3.01 | 17.53 [a,b] ± 1.85 |
| ZS × Du | 61.64 [a] ± 1.26 | 18.85 [a] ± 3.14 | 19.51 [a] ± 2.09 |
| (ZS × ZS) × Du | 65.06 [a] ± 7.58 | 21.36 [a] ± 1.62 | 13.58 [b] ± 7.49 |
| ZS × PLW | 70.23 [b] ± 1.58 | 14.88 [b] ± 1.04 | 14.89 [b] ± 0.81 |
| *p*-value | 0.005 | 0.0003 | 0.038 |
| Average for genetic groups and sex | 62.12 ± 6.25 | 19.69 ± 3.13 | 18.27 ± 4.74 |

x ± SD—mean and standard deviation, mean values in the same columns designated by the different letters differ significantly at: [a,b,c,d]—$p \leq 0.05$, ZS—Złotnicka Spotted × Złotnicka Spotted; ZS × D—Złotnicka Spotted × Duroc; (ZS × ZS) × D—Złotnicka Spotted × Złotnicka Spotted/Duroc; ZS × PLW—Złotnicka, Spotted × Polish Large White.

**Table 3.** Effect of breed or crossing of fatteners on muscle fibre type size (∅ μm) of *m. longissimus thoracis*.

| Genetic Groups of Fatteners | Muscle Fibre Type Percentages | | |
|---|---|---|---|
| | IIB $\bar{x} \pm$ SD | IIA $\bar{x} \pm$ SD | I $\bar{x} \pm$ SD |
| Breed | | | |
| ZS | 64.71 [a] ± 5.97 | 52.58 [a] ± 4.08 | 57.11 [a] ± 4.54 |
| ZS × D | 59.99 [b,c] ± 2.21 | 52.78 [a] ± 4.28 | 50.51 [b] ± 2.31 |
| (ZS × ZS) × D | 61.76 [a,b] ± 3.46 | 49.96 [a,c] ± 2.72 | 56.49 [a] ± 4.79 |
| ZS × PLW | 57.22 [c] ± 3.49 | 48.09 [b,c] ± 2.20 | 52.56 [b] ± 1.90 |
| *p*-value | 0.007 | 0.0000 | 0.0001 |
| Barrows | | | |
| ZS | 66.62 [a] ± 5.43 | 52.78 [a] ± 4.28 | 57.42 [a] ± 4.66 |
| ZS × D | 60.36 [b] ± 1.41 | 46.96 [b] ± 2.07 | 50.71 [b] ± 1.63 |
| (ZS × ZS) × D | 61.27 [a,b] ± 4.28 | 49.34 [a,b] ± 1.44 | 55.22 [a,b] ± 5.20 |
| ZS × PLW | 56.26 [b] ± 4.47 | 47.06 [b] ± 2.43 | 52.71 [a,b] ± 2.36 |
| *p*-value | 0.0001 | 0.004 | 0.032 |
| Gilts | | | |
| ZS | 62.79 ± 6.13 | 52.38 [a] ± 4.09 | 56.78 [a,b] ± 4.64 |
| ZS × D | 59.61 ± 2.94 | 44.17 [b] ± 2.05 | 50.32 [c] ± 3.04 |
| (ZS × ZS) × D | 62.25 ± 2.85 | 50.57 [a] ± 3.70 | 57.56 [a] ± 4.54 |
| ZS × PLW | 58.19 ± 2.25 | 49.11 [a] ± 1.54 | 52.41 [b,c] ± 1.60 |
| *p*-value | 0.863 | 0.0000 | 0.0001 |
| Average for genetic groups and sex | 62.12 ± 6.25 | 19.69 ± 3.13 | 18.27 ± 4.74 |

x ± SD—mean and standard deviation, mean values in the same columns designated by the different letters differ significantly at: [a,b,c]—$p \leq 0.05$, ZS—Złotnicka Spotted × Złotnicka Spotted; ZS × D—Złotnicka Spotted × Duroc; (ZS × ZS) × D—Złotnicka Spotted × Złotnicka Spotted/Duroc; ZS × PLW—Złotnicka Spotted × Polish Large White.

In the case of experiments conducted by Bogucka and Kapelański [48], the diameter of IIB fibres was 57.43 μm, for IIA fibres—41.56 μm and for Type I—47.85 μm. In our studies, the diameter of muscle fibres derived from the *longissimus thoracis* muscle of the Złotnicka Spotted breed (ZS × ZS) amounted to 64.71 μm (IIB), 52.58 μm (IIA) and 57.11 μm (I) and was greater in comparison with hybrids of Duroc and Polish Large White breeds as shown in Table 3. The crossing of Złotnicka Spotted with Duroc and Polish Large White breeds led to a significant decrease of the muscle fibre diameter in hybrids. Oksbjerg et al. [25] reported that the increased diameter of muscle fibres in heavy fatteners exerted a favourable impact on meat processing properties because muscles made up of thicker fibres are more sensitive to massaging treatment. However, according to Bogucka and Kapelański [49], a smaller thickness of the fibres favourably affects meat quality, and it might be considered an indicator of a delicate structure of the meat. Analysing profiles and diameters of muscle fibres of 180-day old gilts of Polish Large White, Polish Landrace, Duroc, Pietrain and Puławska breeds, Wojtysiak et al. [50] found that Puławska breed fatteners exhibited a tendency towards a smaller percentage share of type IIB muscle fibres in the *longissimus lumborum* muscle. Investigations carried out so far by various researchers proved that muscle fibre type composition (%) and fibre type size (∅) were specific for different breeds and lines [51,52]. There is no consensus in the literature on the subject of differences in muscle fibre composition among breeds. Henckel et al. [24] reported that the percentage of fibre types was similar between Landrace and Large White pigs. Essen-Gustavsson and Fjelkner Modig [53] noted that muscle fibre composition was similar in Swedish Landrace, Yorkshire and Hampshire. On the other hand, Chang et al. [54] found that Large White had the lowest and Duroc the highest amount of myosin heavy chain slow fibres in their *longissimus* muscle and *psoas* muscles. However, with respect to the amount of type IIB(IIX) fibres, there were no significant differences between Berkshire, Duroc and Tamworth pigs. In a recent study, Ryu et al. [55] pointed out that in Berkshire pigs, the area of type I fibres

was higher and that of type IIB fibres lower than that of Landrace pigs. Ruusunen and Puolanne [28], who investigated the muscles of Hampshire, Landrace and Yorkshire pigs, found that the muscles of Hampshire pigs were more oxidative compared to those of the others pigs, and showed that variations in muscle fibre composition in pigs within the breeds were larger than the average variations between the breeds. Muscle fibre composition is influenced by age/body weight and the growth rate of animals [33,56,57]. Many researchers have examined the relationship between muscle fibre type characteristics and meat quality [51,58–60]. Kłosowska [51] conducted experiments on pigs of the Złotnicka Spotted and the Pietrain breeds as well as their hybrids and found that smaller proportions of muscles of fast glycolytic character (IIB fibres) were associated with more advantageous meat quality features. Kasprzyk and Bogucka [61] compared the quality of the meat of the native Puławska breed with the hybrids of fattening pigs and found better biological properties of Puławska meat than DanB and Naima hybrids, which is explained by the higher number of slow-twitch oxidative (Type I) and lower number of fast-twitch glycolytic (Type IIA) muscle fibres, which are found in the muscles of the Puławska pig. The higher content of slow-twitch oxidative (Type I) fibres noted in the native breed is a characteristic feature of wild boar and primitive breeds [32,62].

Table 4 shows the quality characters of raw loin (*m. longissimus thoraci*) of the different genotypes. We found differences for a* parameter and no differences for cooking loss (%). These results are consistent with the data of Florowski et al. [37], who reported significantly lower drip loss and WHC values for the *m. longissimus thoracis* of ZS pigs compared to Puławska and Polish Landrace pigs. Additionally, earlier studies showed lower drip losses in Korean native black pigs [63] and higher $pH_u$ values in Iberian pigs [64] for *m. longissimus lumborum* compared with commercial breeds. Variations in the fibre type composition may affect meat colour. The right colour of meat can be conditioned by the ferrous oxymioglobin (oxyMb) [65], which is directly connected with the percentage and the size of muscle fibre types [66]. Henckel et al. [24] suggested that haem pigments had a positive correlation with the percentage of type I muscle fibres and a negative correlation with that of type IIB fibres. The present research showed that the a* color parameter was positively correlated with the frequency of the muscle fibers IIA and the size of all types of muscle fibers (Ø µm). The above correlations were significant, as shown in Table 5. Colour parameters (L*, and a* values) of the *m. longissimus thoracis* of ZS pigs and the hybrids of this breed correspond to the meat parameters of other native pig breeds as reported [64,67]. According to Bogucka and Kapelański [49], the meat of Złotnicka spotted pigs compared to the Puławska breed was darker, which was associated with a greater percentage of type I fibres and a smaller percentage of type IIB fibres. The colour of meat and the fat content may also be a significant determinant in the evaluation of pork quality [68]. Hybrid (ZS × ZS) × D pigs were characterised by a significantly higher intramuscular fat content (4.08%) in the *longissimus thoracis* muscle in comparison with ZS × PLW hybrid pigs (2.95%). Bejerholm and Barton-Gade [69] proved that the intramuscular fat (IMF) between 2% and 3% is to be considered as an optimal level. The IMF is an important marker of meat quality because higher contents of intramuscular fat exert a positive influence on sensory characteristics and the technological and culinary usefulness of meat [5,70,71]. The IMF is of particular importance in the case of raw meat used for ripening products. It is an element enabling a normal course of ripening processes and a factor responsible for the favourable development of sensory characteristics of such products [72,73]. It is generally accepted that local breeds produce a higher IMF content [1,37,63,64,74]. As Serrano et al. [75] indicate, muscles of the Spanish breed Iberico are characterised by a particularly high content of fat (8.8%). Considerable IMF ranging from 3.32 to 4.27% in the *m. longissimus dorsi* was reported for the native Italian breed of Nero Siciliano [76]. On the other hand, Čandek-Potokar et al. [77] determined the content of IMF in the Slovenian breed of Krškopolje at 3%. The results of research on the IMF in Polish native pigs vary. Rak et al. [78] determined the content of IMF in ZS pigs at 1.02–2.07%. Grześkowiak et al. [79] noted the mean content of IMF of 2.04% in Złotnicka Spotted pigs and 1.87% in the Złotnicka White breed. However,

the IMF content in ZS pigs observed in the research by Florowski et al. [80] was higher and amounted to 3.1%. In the present research, the IMF content in ZS × ZS porkers was 3.44%. The highest IMF content of 4.08% was noted for the porkers from the (ZS × ZS) × D group. The animals from the ZS × PLW group were characterised by the smallest fat content reaching the value of 2.95%. The differences observed between the (ZS × ZS) × D and ZS × PLW groups were significant ($p \leq 0.05$). Lefaucheur [81] stated that there is no universal relationship between muscle fibre composition and IMF content in meat, and they assume that both these traits are rather independent.

**Table 4.** Quality characteristics of raw meat *m. longissimus thoracis*.

| Genetic Groups of Fatteners | Colour Parameters | IMF (%) | Protein (%) | pH | | |
|---|---|---|---|---|---|---|
| | L* | a* | b* | | | |
| | | | Breed | | | |
| ZS | 46.43 ± 5.46 | 8.17 [a] ± 0.99 | 2.89 ± 1.88 | 3.44 [a,b] ± 0.92 | 24.54 ± 1.42 | 6.39 ± 0.31 |
| ZS × D | 47.33 ± 4.15 | 6.87 [b] ± 1.54 | 2.93 ± 1.80 | 3.52 [a,b] ± 0.76 | 24.19 ± 1.11 | 6.38 ± 0.25 |
| (ZS × ZS) × D | 46.05 ± 6.71 | 7.98 [a] ± 1.46 | 2.86 ± 1.65 | 3.80 [a] ± 1.54 | 25.41 ± 1.07 | 6.26 ± 0.36 |
| ZS × PLW | 47.12 ± 1.74 | 6.67 [b] ± 0.70 | 2.17 ± 0.62 | 2.95 [b] ± 0.32 | 24.32 ± 1.24 | 6.42 ± 0.26 |
| *p*-value | 0.085 | 0.005 | 0.168 | 0.007 | 0.168 | 0.768 |
| | | | Barrows | | | |
| ZS | 46.46 ± 6.33 | 8.01 ± 0.80 | 2.63 [a,b] ± 2.12 | 3.65 [a,b] ± 1.03 | 24.43 ± 1.34 | 6.37 ± 0.23 |
| ZS × D | 47.75 ± 5.12 | 7.34 ± 2.02 | 3.56 [a] ± 2.69 | 3.51 [b] ± 0.89 | 23.77 ± 0.77 | 6.26 ± 0.17 |
| (ZS × ZS) × D | 41.39 ± 3.19 | 7.14 ± 0.70 | 0.72 [b] ± 1.11 | 4.85 [a] ± 1.16 | 25.65 ± 0.93 | 6.51 ± 0.18 |
| ZS × PLW | 47.42 ± 1.70 | 6.58 ± 0.82 | 2.07 [a,b] ± 0.40 | 2.88 [b] ± 0.36 | 23.57 ± 0.75 | 6.41 ± 0.12 |
| *p*-value | 0.384 | 0.0000 | 0.16 | 0.0051 | 0.658 | 0.248 |
| | | | Gilts | | | |
| ZS | 46.11 ± 4.78 | 8.34 [a] ± 1.17 | 3.14 [a,b] ± 1.70 | 2.95 [a,b] ± 0.67 | 24.65 ± 1.56 | 6.40 ± 0.38 |
| ZS × D | 46.91 ± 3.48 | 6.42 [b] ± 0.87 | 2.31 [b] ± 0.94 | 3.53 [b] ± 0.70 | 24.60 ± 1.33 | 6.50 ± 0.27 |
| (ZS × ZS) × D | 50.71 ± 6.07 | 8.84 [a] ± 1.58 | 4.99 [a] ± 3.14 | 3.31 [a] ± 1.59 | 25.73 ± 1.12 | 6.01 ± 0.32 |
| ZS × PLW | 46.83 ± 1.72 | 6.76 [b] ± 0.55 | 2.27 [b] ± 0.77 | 3.02 [b] ± 0.27 | 25.08 ± 1.03 | 6.41 ± 0.37 |
| *p*-value | 0.501 | 0.0008 | 0.037 | 0.001 | 0.08 | 0.105 |
| Average for genetic groups and sex | 46.67 ± 4.87 | 7.58 ± 1.32 | 2.75 ± 2.03 | 3.43 ± 1.02 | 24.66 ± 1.31 | 24.66 ± 1.31 |

x ± SD—mean and standard deviation, mean values in the same columns designated by the different letters differ significantly at: [a,b]—$p \leq 0.05$, ZS—Złotnicka Spotted × Złotnicka Spotted; ZS × D—Złotnicka Spotted × Duroc; (ZS × ZS) × D—Złotnicka Spotted × Złotnicka Spotted/Duroc; ZS × PLW—Złotnicka Spotted × Polish Large White.

**Table 5.** Correlation coefficients I between quality characteristics of raw meat *m. longissimus thoracis* and muscle fibre type percentage and muscle fibre type size.

| | Muscle Fibre Type | | |
|---|---|---|---|
| | IIB | IIA | I |
| | Muscle fibre type percentages % | | |
| Colour parameter L | 0.114 (0.651) | −0.062 (0.067) | −0.102 (0.581) |
| Colour parameter a* | −0.222 (0.098) | **0.350 (0.0301)** | 0.087 (0.089) |
| Colour parameter b* | 0.020 (0.89) | 0.086 (0.541) | −0.038 (0.556) |
| Intramuscular fat (%) | −0.244 (0.072) | −0.199 (0.087) | 0.186 (0.406) |
| Protein content (%) | 0.010 (0.308) | 0.214 (0.922) | −0.175 (0.178) |
| pH$_{45}$ | 0.039 (0.409) | −0.246 (0.0881) | 0.094 (0.519) |
| | Muscle fibre type size (Ø μm) | | |
| Colour parameter L* | 0.034 (0.081) | 0.081 (0.521) | 0.155 (0.755) |
| Colour parameter a* | **0.314 (0.019)** | **0.464 (0.0001)** | 0.382 (0.0079) |
| Colour parameter b* | 0.098 (0.0671) | 0.213 (0.844) | 0.275 (0.688) |
| Intramuscular fat IMF (%) | 0.121 (0.177) | −0.139 (0.511) | 0.017 (0.911) |
| Total protein content (%) | 0.069 (0.877) | 0.058 (0.651) | 0.190 (0.077) |
| pH$_{45}$ | 0.070 (0.755) | −0.055 (0.677) | −0.216 (0.114) |

Correlation values with *p*-value in bracket.

The meat of the Złotnicka Spotted breed of pigs was characterised by good quality, with an appropriate dark colour and an optimal IMF content level. A similar conclusion was drawn by Jankowiak et al. [82]. Hybrid (ZS × ZS) × D pigs were characterised by the highest IMF content in the *longissimus thoracis* muscle resulting in the lowest cutting force of roasted loin and the lowest thermal drip (Table 6).

**Table 6.** Quality characteristics of the roasted loin *m. longissimus thoracis*.

| Genetic Groups of Fatteners | Hardness [N] | Springiness | Cohesiveness | Chewiness [N] | Resilience | Warner–Bratzler Shear Force [kG/cm$^2$] | Cooking Loss % |
|---|---|---|---|---|---|---|---|
| | | | | Breed | | | |
| ZS | 146.71 ± 41.62 | 0.48 ± 0.05 | 0.50 ± 0.05 | 38.42 ± 14.63 | 0.19 ± 0.03 | 6.13 ± 1.95 | 30.12 ± 6.47 |
| ZS × D | 143.32 ± 32.98 | 0.48 ± 0.04 | 0.48 ± 0.04 | 33.60 ± 11.12 | 0.18 ± 0.02 | 6.31 ± 1.61 | 29.16 ± 7.44 |
| (ZS × ZS) × D | 141.37 ± 45.29 | 0.61 ± 0.07 | 0.5 ± 0.05 | 37.93 ± 16.84 | 0.18 ± 0.02 | 4.63 ± 0.65 | 27.54 ± 8.77 |
| ZS × PLW | 130.99 ± 39.83 | 0.45 ± 0.04 | 0.50 ± 0.05 | 30.03 ± 10.33 | 0.19 ± 0.02 | 5.95 ± 1.59 | 30.42 ± 6.52 |
| *p*-value | 0.85 | 0.511 | 0.098 | 0.083 | 0.56 | 0.266 | 0.061 |
| | | | | Barrows | | | |
| ZS | 138.88 ± 40.21 | 0.46 ± 0.05 | 0.49 ± 0.07 | 34.20 ± 12.55 | 0.19 ± 0.03 | 5.85 ± 1.65 | 30.68 ± 5.69 |
| ZS × D | 143.46 ± 7.92 | 0.48 ± 0.05 | 0.46 ± 0.04 | 31.09 ± 3.52 | 0.17 ± 0.01 | 6.17 ± 1.41 | 30.66 ± 3.97 |
| (ZS × ZS) × D | 107.56 ± 34.08 | 0.46 ± 0.05 | 0.5 ± 0.04 | 24.9 ± 9.67 | 0.18 ± 0.02 | 4.43 ± 0.30 | 23.35 ± 11.03 |
| ZS × PLW | 128.62 ± 35.36 | 0.45 ± 0.04 | 0.47 ± 0.03 | 27.67 ± 10.06 | 0.19 ± 0.02 | 6.27 ± 1.92 | 28.35 ± 7.48 |
| *p*-value | 0.07 | 0.732 | 0.168 | 0.073 | 0.945 | 0.851 | 0.671 |
| | | | | Gilts | | | |
| ZS | 154.54 ± 43.63 | 0.51 [a,b] ± 0.05 | 0.52 ± 0.031 | 42.62 ± 15.97 | 0.2 ± 0.02 | 6.41 ± 2.26 | 29.55 ± 7.45 |
| ZS × D | 143.18 ± 48.83 | 0.48 [b] ± 0.03 | 0.50 ± 0.024 | 36.11 ± 15.81 | 0.18 ± 0.12 | 6.56 ± 1.96 | 27.65 ± 10.16 |
| (ZS × ZS) × D | 175.17 ± 24.43 | 0.55 [a] ± 0.04 | 0.49 ± 0.057 | 50.97 ± 10.96 | 0.18 ± 0.02 | 4.85 ± 0.86 | 31.68 ± 2.89 |
| ZS × PLW | 133.37 ± 48.01 | 0.46 [b] ± 0.05 | 0.53 ± 0.044 | 32.39 ± 11.17 | 0.19 ± 0.02 | 5.64 ± 1.32 | 32.49 ± 5.38 |
| *p*-value | 0.276 | 0.005 | 0.168 | 0.07 | 0.1961 | 0.109 | 0.668 |
| Average for genetic groups | 141.82 ± 39.67 | 0.48 ± 0.05 | 0.5 ± 0.05 | 35.68 ± 13.75 | 0.19 ± 0.02 | 5.83 ± 1.7 | 29.47 ± 7.03 |

x ± SD—mean and standard deviation, mean values in the same columns designated by the different letters differ significantly at: [a,b]—$p \leq 0.05$. ZS—Złotnicka Spotted × Złotnicka Spotted; ZS × D—Złotnicka Spotted × Duroc; (ZS × ZS) × D—Złotnicka Spotted × Złotnicka Spotted/Duroc; ZS × PLW—Złotnicka Spotted × Polish Large White.

Table 7 presents quality traits of raw smoked loin derived from the *longissimus* muscle of Złotnicka Spotted breed pigs and hybrids. The best tenderness was recorded in the raw smoked loin derived from the *longissimus* muscle of Złotnicka Spotted breed pigs (20.99 N), while tenderness of the same loin from the *longissimus* muscle of ZS × D and ZS × PLW hybrids was worse (from 36.08 to 43. 72 N). This could have been caused by the highest proportions of fibres in the *longissimus thoracis* muscle of pigs of Złotnicka Spotted breed. Essen-Gustavsson et al. [17] demonstrated that fibres were the main location of the IMF deposition. Therefore, it is possible that the higher percentage proportions of muscle fibres observed in the muscles of the Złotnicka Spotted breed could have effectively improved loin tenderness. Moreover, Cameron et al. [16] found that the increased surface area of white muscle fibres correlated positively with tenderness and negatively with juiciness. On the other hand, the surface area of muscle fibres correlates positively with juiciness and negatively with tenderness. The surface increase of muscle fibres may be associated with the daily body weight gains [25,59]. Studies in pigs showed that the frequency of fast glycolytic fibres (white fibres—type IIB) was negatively correlated with toughness [83]. Other studies suggest that differences between the same muscles may only be evident in specific breeds of any one species. It was shown that the sheer force of the *longissimus* muscle differs significantly in Duroc but not in Berkshire or Large White pigs [54]. The relationship between some quality traits (e.g., textural parameters) and muscle fibre characteristics remain unclear [84].

**Table 7.** Quality characteristics of the raw smoked loin ($\bar{x}$, $\pm$).

| Genetic Groups of Fatteners | Colour Parameters | | | IMF (%) | Protein (%) | pH | Tenderness (N) |
|---|---|---|---|---|---|---|---|
| | L* | a* | b* | | | | |
| | | | | Breed | | | |
| ZS | 49.38 ± 5.58 | 7.64 [a] ± 1.22 | 4.51 ± 2.68 | 3.02 [a] ± 1.01 | 23.59 [a] ± 0.93 | 5.59 ± 0.053 | 20.99 [a] ± 7.63 |
| ZS × D | 52.46 ± 4.87 | 7.24 [a] ± 1.26 | 4.03 ± 1.72 | 3.23 [ab] ± 1.22 | 21.76 [b,c] ± 1.28 | 5.67 ± 0.06 | 36.08 [b] ± 6.88 |
| (ZS × ZS) × D | 50.52 ± 3.82 | 7.42 [a] ± 1.99 | 4.08 ± 1.90 | 4.17 [b] ± 1.73 | 22.19 [b] ± 0.80 | 5.65 ± 0.10 | 38.83 [b,c] ± 6.74 |
| ZS × PLW | 49.42 ± 3.14 | 5.14 [b] ± 1.18 | 2.56 ± 1.64 | 1.89 [c] ± 0.48 | 21.36 [c] ± 0.61 | 5.63 ± 0.03 | 43.72 [c,b] ± 8.02 |
| *p*-value | 0.088 | 0.0011 | 0.08 | 0.003 | 0.000 | 0.296 | 0.0061 |
| | | | | Barrows | | | |
| ZS | 50.24 ± 6.43 | 7.53 ± 1.09 | 5.13 ± 3.03 | 3.28 ± 1.06 | 23.43 [a] ± 0.82 | 5.59 ± 0.67 | 20.21 [a] ± 8.05 |
| ZS × D | 53.56 ± 3.89 | 7.53 ± 1.54 | 4.55 ± 1.39 | 3.79 ± 1.34 | 21.17 [b] ± 0.71 | 5.68 ± 0.06 | 40.19 [b] ± 7.34 |
| (ZS × ZS) × D | 48.97 ± 4.69 | 6.98 ± 0.78 | 2.58 ± 1.80 | 4.55 ± 1.99 | 21.79 [b] ± 0.99 | 5.71 ± 0.11 | 34.71 [b] ± 5.08 |
| ZS × PLW | 50.34 ± 4.01 | 5.30 ± 1.29 | 2.68 ± 2.07 | 1.98 ± 0.65 | 21.36 [b] ± 0.53 | 5.62 ± 0.03 | 38.63 [b] ± 3.84 |
| *p*-value | 0.0871 | 0.781 | 0.0981 | 0.491 | 0.000 | 0.0673 | 0.001 |
| | | | | Gilts | | | |
| ZS | 48.51 ± 4.77 | 7.75 [a] ± 1.38 | 3.97 ± 2.28 | 2.76 [a,b] ± 0.93 | 23.76 ± 1.04 | 5.58 ± 0.04 | 21.76 [a] ± 7.53 |
| ZS × D | 51.37 ± 5.94 | 6.96 [a,b] ± 1.01 | 3.51 ± 2.01 | 2.68 [a,b] ± 0.87 | 22.36 ± 1.52 | 5.66 ± 0.07 | 31.96 [a,b] ± 3.22 |
| (ZS × ZS) × D | 52.07 ± 2.18 | 7.87 [a] ± 2.81 | 5.38 ± 1.30 | 3.78 [a,b] ± 1.55 | 22.58 ± 0.27 | 5.59 ± 0.06 | 42.94 [c] ± 5.82 |
| ZS × PLW | 48.31 ± 1.73 | 4.97 [b] ± 1.17 | 2.47 ± 1.32 | 1.80 [a,b] ± 0.29 | 21.36 ± 0.75 | 5.63 ± 0.04 | 48.82 [c] ± 8.07 |
| *p*-value | 0.451 | 0.0002 | 0.781 | 0.0003 | 0.0881 | 0.0981 | 0.000 |
| Average for genetic groups | 50.23 ± 4.73 | 7.02 ± 1.67 | 3.93 ± 2.27 | 3.07 ± 1.34 | 22.50 ± 1.31 | 5.62 ± 0.07 | 32.12 ± 11.91 |

x ± SD—mean and standard deviation, mean values in the same columns designated by the different letters differ significantly at: [a,b,c]—$p \leq 0.05$, ZS—Złotnicka Spotted × Złotnicka Spotted; ZS × D—Złotnicka Spotted × Duroc; (ZS × ZS) × D—Złotnicka Spotted × Złotnicka Spotted/Duroc; ZS × PLW—Złotnicka Spotted × Polish Large White.

Table 8 presents correlation coefficients between quality parameters of roasted loin and the percentage share and diameter of individual types of muscle fibres of all genotypes. These correlations were insignificant. Worse correlations were found between the percentage share and individual types of muscle fibres and tenderness, chewability, shear force and the thermal drip of roasted loin. ZS × ZS. Table 9 presents the correlation coefficients between the quality parameters of the raw smoked loin and the percentage share and diameter of individual types of muscle fibres of all genotypes. Some correlations were statistically significant. The IIB muscle fibre percentages were negatively correlated with color parameters a* and b*, % IMF and protein content, while the IIA muscle fibres percentages was positively correlated with colour parameters a*, % IMF and protein content. Tenderness was highly positively correlated with % IIB muscle fibre (0.593) and negatively correlated with % IIA muscle fibres and % I muscle fibres ($-0.522$ and $-0.423$, respectively). The raw smoked loin ZS × PLW hybrid pigs had poorer tenderness (43.72 N) compared to the Złotnicka Spotted breed pigs (20.99 N). Muscle fibre type size was positively correlated with colour parameters a* and b* while protein content was negatively correlated with pH$_{45}$ and tenderness. Złotnicka Spotted breed pigs were characterized by the greatest diameter of muscle fibers (the best tenderness of raw smoked loin) while ZS × PLW hybrid pigs were characterized by the smallest diameter of muscle fibers (the worst tenderness of raw smoked loin). (ZS × ZS) × D hybrid pigs had a similar diameter of muscle fibers as Złotnicka Spotted breed pigs. Quality parameters of raw, roasted loin and raw smoked loin from the ZS × ZS × Duroc hybrid pigs were the closest to the quality parameters of Złotnicka Spotted breed pigs. The study results of Szulc et al. [13] indicate that crossing of ZS and Duroc breeds improves fattening and slaughter performance, while maintaining good meat quality in their crosses. The crosses of both the breeds kept in extensive breeding may be successfully used in high quality meat production. The pork from such animals may be a raw material for production of regional meat products [6]. The use of meat from crosses in meat processing may improve both the quality of the processed products and the efficiency of production based on the native Złotnicka Spotted breed.

**Table 8.** Correlation coefficients[®] between the quality characteristics of the roasted loin and muscle fibre percentage and fibre type size.

| Quality Characteristics | Muscle Fibre Type | | |
| --- | --- | --- | --- |
| | IIB | IIA | I |
| Muscle fibre type percentages % | | | |
| Hardness (N) | 0.017 (0.182) | 0.033 (0.288) | −0.044 (0.088) |
| Springiness | 0.067 (0.6456) | 0.193 (0.1780) | −0.240 (0.0943) |
| Cohesiveness | 0.075 (0.6023) | −0.012 (0.9337) | −0.085 (0.5580) |
| Chewiness (N) | −0.002 (0.9887) | 0.121 (0.4007) | −0.089 (0.5375) |
| Resilience | −0.031 (0.8308) | 0.009 (0.9512) | 0.026 (0.8564) |
| Shear force (kG/cm$^2$) | −0.049 (0.165) | 0.091 (0.0861) | 0.011 (0.0691) |
| Cooking loss (%) | 0.040 (0.8811) | −0.069 (0.5591) | 0.001 (0.5814) |
| Muscle fibre type size (Ø μm) | | | |
| Hardness (N) | 0.144 (0.671) | −0.035 (0.711) | −0.001 (0.911) |
| Springiness | −0.015 (0.9157) | −0.039 (0.7857) | −0.049 (0.7339) |
| Cohesiveness | 0.081 (0.5748) | 0.077 (0.5931) | 0.019 (0.8968) |
| Chewiness (N) | 0.112 (0.4363) | 0.016 (0.9109) | 0.044 (0.7605) |
| Resilience | 0.254 (0.0743) | 0.194 (0.1771) | 0.154 (0.2857) |
| Shear force (kG/cm$^2$) | 0.019 (0.8968) | 0.072 (0.6203) | −0.028 (0.8446) |
| Cooking loss (%) | 0.096 (0.054) | −0.044 (0.7611) | −0.076 (0.5994) |

Correlation values with *p*-value in bracket.

**Table 9.** Correlation coefficients[®] between the quality characteristics of the raw smoked loin and muscle fibre percentage and fibre type size.

| Quality Characteristics | Muscle Fibre Type | | |
| --- | --- | --- | --- |
| | IIB | IIA | I |
| Muscle fibre type percentages % | | | |
| Colour parameter L* | 0.014 (0.077) | 0.038 (0.551) | −0.047 (0.299) |
| Colour parameter a* | **−0.342 (0.027)** | **0.415 (0.0002)** | 0.195 (0.343) |
| Colour parameter b* | −0.227 (0.2881) | 0.254 (0.5114) | 0.126 (0.411) |
| Intramuscular fat IMF (%) | **−0.287 (0.018)** | **0.290 (0.027)** | 0.206 (0.171) |
| Total protein content (%) | **−0.471 (0.008)** | **0.574 (0.0000)** | 0.213 (0.331) |
| pH$_{45}$ | 0.017 (0.117) | −0.042 (0.099) | 0.017 (0.511) |
| Tenderness (N) | **0.593 (0.009)** | **−0.522 (0.039)** | **−0.423 (0.0021)** |
| Muscle fibre type size (Ø μm) | | | |
| Colour parameter L* | −0.002 (0.085) | 0.018 (0.855) | −0.082 (0.511) |
| Colour parameter a* | 0.263 (0.441) | **0.307 (0.022)** | **0.332 (0.422)** |
| Colour parameter b* | 0.133 (0.688) | 0.155 (0.581) | 0.164 (0.499) |
| Intramuscular fat (%) | 0.204 (0.499) | −0.121 (0.229) | 0.092 (0.611) |
| Protein content (%) | **0.402 (0.0027)** | **0.408 (0.0000)** | **0.341 (0.039)** |
| pH$_{45}$ | −0.073 (0.339) | −0.341 (0.491) | −0.244 (0.088) |
| Tenderness (N) | **−0.347 (0.045)** | **−0.310 (0.013)** | −0.132 (0.078 |

Correlation values with *p*-value in bracket.

## 4. Conclusions

In Polish conditions there are possibilities to produce raw pork on the basis of some native breeds, e.g., Złotnicka Spotted or their hybrids with other breeds, with the meat being useful for ripening products. In order to prevent the loss of precious qualitative values of the meat of porkers from crossbreeding of native breeds, it is very important to select appropriate breeds. The results of earlier studies provide conclusive evidence that Złotnicka Spotted pigs can be used in breeding programs aimed at producing valuable slaughter product, which meets high technological requirements. Bearing in mind the small numbers of Złotnicka Spotted breeding pigs, the only way to produce appropriate quantities of fatteners of good meat quality parameters suitable for manufacturing raw ripening articles is to cross pigs of this breed with high population breeds. It seems that

crossing the Złotnicka Spotted breed with Duroc pigs is likely the most appropriate solution. In order to support the commercial crossing of pigs of this breed with other meat breeds, it is essential to realize the assumptions of the project of the "Conservation program of the Złotnicka Spotted breed genetic resources" and to further increase the number of animals of this breed.

**Author Contributions:** Conceptualization, K.S.; Methodology, K.S. and W.M.; Data collection, K.S., D.W., W.M. and Ł.M.; Investigation, K.S., D.W., W.M. and Ł.M.; Data Curation, K.S. and W.M.; Writing—Original Draft Preparation, K.S., D.W., W.M. and Ł.M.; Writing—Review & Editing, W.M. and Ł.M.; Visualization, Ł.M. All authors have read and agreed to the published version of the manuscript.

**Funding:** This research was funded by Polish Ministry of Scientific Research and Information Technology grant number PBZ-KBN-N N311 266336.

**Institutional Review Board Statement:** This study did not require ethical approval because all the procedures were breeding procedures. Slaughter was conducted at the authorized slaughterhouse, following the normal commercial procedure.

**Informed Consent Statement:** Not applicable.

**Data Availability Statement:** Data available on request due to restrictions e.g., privacy or ethical.

**Acknowledgments:** This study was funded by the Polish Ministry of Scientific Research and Information Technology grant PBZ-KBN-N N311 266336.

**Conflicts of Interest:** The authors declare no conflict of interest.

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
