# Peer review of "The Muscle Fibre Characteristics and the Meat Quality of m. longissimus thoracis from Polish Native Złotnicka Spotted Pigs and the Crossbreed Fatteners from the Crossing of Duroc and Polish Large White Boars"

_applsci, doi:10.3390/app12063051_

Round 1

Reviewer 1 Report

Please, find the review in the attached file.

Author Response

Dear Reviewers

Please find corrected version of manuscript. As many suggestion were same or similar here we answered all doubts/misspelling/suggestion from both reviewers and merge answers into one file 

Reviewer 2 Report

While there are  good reasons for the evaluation of meat from the native breed crosses the authors should avoid using non-significant meat quality data to support their underlying keenness to highlight the value of the polish native breed. There is sufficient clear significant data to do this

There are a range of changes and corrections suggested in the attachment

Author Response

Dear Reviewer,

Please find corrected version of manuscript. As many suggestion were same or similar here we answered all doubts/misspelling/suggestion from both reviewers and merge answers into one file 

Round 2

Reviewer 1 Report

The authors provide a new version without highlighted changes. This makes it very difficult to review  the new version.
Also, in the joint Rebuttal letter to the two reviewers, I don't see any reference to my review. Especially with regard to Major / Modrate concerns ... This reviewer has already discussed these facts with the Editor. Maybe there was a mistake in generating the rebuttal letter?

Author Response

Dear Reveiwer, 

I'm very sorry - There  were some problem with uploading file and I did not realized before accepting that answears to Your review were not added.

Please find them in file below - as I said because most comments were similar I created one file for both reviewers with answears 

Reviewer 2 Report

The manuscript is much improved but there remain some points that need to be considered and attended too.

Ln 1: remove ‘performed investigations was’ add ‘investigations were’
Ln 9: remove ‘comparison’ add ‘compared’

Ln 16: remove ‘smallest’ add ‘lowest’

Ln 17: ‘toother’ problem with the text’
Ln 18: remove ‘characterised by the best tenderness’ add ‘the most tender’

Ln 19: remove ‘hybrids of the Złotnicka Spotted breed with Duroc and Polish Large White breeds was worse’ add ‘of the Złotnicka Spotted bred hybrids with the Duroc and Polish Large White breeds was poorer’

Ln 23: remove ‘Duroc breed pigs is a good solution.’ Add ‘Duroc pigs would be a suitable solution.’

Ln 118: remove ‘ageing’ duplicated in sentence.

Ln 138: , a* > 0 , a* < 0 green, b*’ note what colour a* > 0 , trends to red

Ln 244: remove ‘fat’ add ‘far’

Ln 244: remove ‘proved’ add ‘indicate’ as the statement ‘There is no consensus in the literature on the subject about differences in muscle fibre composition’ indicates it has not been proved as an absolute

Ln 257: remove ‘comparing’ add ‘compared’

Ln 264: ‘muscles of glycolytic character of metabolic changes’ this is confusing

Ln 278: remove ‘comparison’ add “compared’

Ln 283: remove ‘Own’ add ‘The present’

Ln 278: remove ‘by’

Ln 311: , In the authors’ own research, the intramuscular fat content in ZS x ZS porkers was 3.30%., which research is this? If previous it needs to be referenced

Ln 352: remove ‘swine groups’ use ‘genotypes’ as in Ln 349.

Ln 358: remove ‘was the worst’ add ‘had poorer’

Ln 360: remove ‘and protein content while’ add while ‘protein content’

Ln 365: remove ‘ZS x duroc’ add ‘from the’
ln 366: remove ‘parameters’ add  ‘parameters of’

Ln 381: remove ‘small’ add ‘of the small’

Ln 384: remove ‘article add ‘articles’

Ln 384: remove ‘characterized by’ having’

Ln 385: remove  ‘of’ add ‘the’
Ln 385: remove ‘is among’ add ‘is likely’

Author Response

Dear reviewer,

As suggested we changed mistakes/unclear parts of manuscript

Best regards

Round 3

Reviewer 1 Report

In my opinion, the content of this third review of the manuscript has improved considerably.
However, the authors should make every effort to review the entire document in order to complete a presentable version. Here are some of the errors / inaccuracies detected:

Abstract: Right justify

Line 258: (Suggestion) I would rephrase and emphazise that the use of this breed is economically suitable

References and elsewhere: Check the size and style types

All Tables: carefully recheck all the details! Examples:

Tables 2 and 3: Should "breed" be changed by "male"?

Table 4: Differente groups and data are not in thesame line

Table 6: P values for Barrows and Gilts?

Author Response

Dear Reviewer,

please find in attachment our responses to Your comments

best regards

Łukasz Migdał
